# An Area-Orientated Analysis of the Temporal Variation of Extreme Daily Rainfall in Great Britain and Australia

**Han Wang** [1,2,3] and **Yunqing Xuan** [3,*]

1   China Institute of Water Resources and Hydropower Research, Beijing 100038, China
2   Research Center on Flood & Drought Disaster Reduction of the Ministry of Water Resources, Beijing 100038, China
3   Department of Civil Engineering, Swansea University Bay Campus, Fabian Way, Swansea SA1 8EN, UK
*   Correspondence: y.xuan@swansea.ac.uk

**Abstract:** This paper presents an analysis of the temporary variation of the area-orientated annual maximum daily rainfall (AMDR) with respect to the three spatial properties: location, size and shape of the region-of-interest (ROI) in Great Britain and Australia using two century-long datasets. The Maximum Likelihood and Bayesian Markov-Chain-Monte-Carlo methods are employed to quantify the time-varying frequency of AMDR, where a large proportion of the ROIs shows a non-decreasing level of most frequent AMDR. While the most frequent AMDR values generally decrease with larger-sized ROIs, their temporal variation that can be attributed to the climate change impact does not show the same dependency on the size. Climate change impact on ROI-orientated extreme rainfall is seen higher for rounded shapes although the ROI shape is not as significant as the other two spatial properties. Comparison of the AMDR at different return levels shows an underestimation by conventionally used stationary models in regions where a nonstationary (i.e., time-varying) model is preferred. The findings suggest an overhaul of the current storm design procedure in view of the impact of not only climate change but also spatial variation in natural processes.

**Keywords:** extreme rainfall; spatial variation; return period; GEV; climate change; nonstationarity

---

## 1. Introduction

Applications of extreme value (EV) theory in modelling meteorological and environmental processes have been widely practised in designing and validating many infrastructure systems [1]. A classical analysis approach adopted in these applications is to use historical hydro-climatic data, such as rainfall, temperature, river flows, etc., to estimate the parameters of the required EV model which would offer probability distributions of the natural phenomenon in question, so as to address its occurrence or exceedance probability at given thresholds. Since Jenkinson [2] proposed a generalized approach to analysing the frequency distribution of annual maxima, many efforts have been made to quantify natural phenomena at extreme levels using the Generalized Extreme Value (GEV) models whose parameters are often fitted by using the Maximum Likelihood (ML) method and L-Moments (LM) method, especially in designing and planning water engineering systems [3–7].

Recently, there has been a growing interest in studying natural events from a climate-change perspective, given that the key hydroclimatic variables, such as precipitation, temperature and streamflow, are indeed changing due to the impact of climate change [8,9]. To address the reliability of infrastructure designs based upon extreme value analysis, stationary risk analyses have been re-assessed from a new adaptive perspective where Sarhadi [10] proposed a multivariate time-varying risk framework for all stochastic multidimensional systems under the influence of a changing environment. For the commonly used nonstationary GEV models, this means that their scale and location parameters can vary

with time or other climate indices [11]. For example, Hasan [12] proposed two nonstationary GEV models for extreme temperature with each model assuming only one parameter as nonstationary depending linearly and exponentially in time, respectively; Sarhadi and Soulis [13] defined both the scale and location parameters for extreme precipitation analysis using a linear, time-varying representation. Their results showed that stationary models tend to have underestimation of extreme precipitation when compared with the nonstationary ones; Panagoulia [14] generated 16 nonstationary GEV models of extreme precipitation with linear time dependence of location and log-linear time dependence of scale, employing the Akaike Information Criterion (AIC) and the Bayesian Information Criterion (BIC) for selecting the best model and examined confidence intervals for model parameters. Unlike the research listed above which assumes a constant shape parameter, Ragulina and Reitan [15] also explored the change in the shape parameter and found that it evidently depends on the altitude of study areas.

Although in the last few decades there have been many studies that apply nonstationary GEV distributions to fit extreme rainfall at different scales, most of them were associated with a limited number of regions of interest (ROIs) within predefined boundaries such as political regions and river catchments [16–19]. It is clear that extreme rainfall can be affected by its local features, not least the topography but also the orientation (shape) and size of the area [20]. However, how the area-orientated rainfall extremes vary with the ROIs' geographical location, size and shape in an context of nonstationarity has not been fully studied; yet it is challenging as the variability of extremes can be sensitive to the size of the regions studied, e.g., substantial trends over smaller regions can arise purely from the natural variability [21–23]. Therefore, this paper presents a comparative study of Great Britain (GB) and Australia (AU) moving forward from our previous study [20], aiming to gain much-needed insights into the spatial variability of extreme rainfall associated with dramatically different climate and geomorphological features (GB and AU), as represented by the nonstationary probability distribution parameters. Not limited by the regional boundaries, a large number of ROIs generated by Wang and Xuan [20] by randomizing their locations, sizes and shapes are reused. The century-long annual maximum daily rainfall (AMDR) time series were extracted with assistance from a high-performance computing (HPC) system. They were then used to fit both stationary and nonstationary GEV models to address the impact of climate change on extreme rainfall. Finally, the patterns changing with three spatial features and the contrasting differences between stationary and nonstationary conditions at different return levels were analysed.

The remainder of this paper is organised as follows: Section 2 describes the ROI-based methods, the data and the trend analysis of extreme rainfall over the last century in two countries. The results are discussed in details focusing on the spatial feature of the stationary and nonstationary GEV models (Section 3.1); spatial changes with ROI sizes (Section 3.2) and shapes (Section 3.3); and the comparison (Section 3.4). The conclusions and recommendations are given in Section 4.

## 2. Methods

The steps below are followed in this study and the details are presented in Figure 1:

- Generate ROIs with varying locations, sizes and shapes and extract the annual maximum daily rainfall (AMDR) time series with the assistance of high-performance computing (HPC).
- Fit the time series obtained at every ROI with stationary and nonstationary GEV models with different parameter estimation methods.
- Evaluate the performance of all models and analyse the changes of time-varying parameters with regard to the geographical locations, sizes, and shapes as well as the level of extremity.

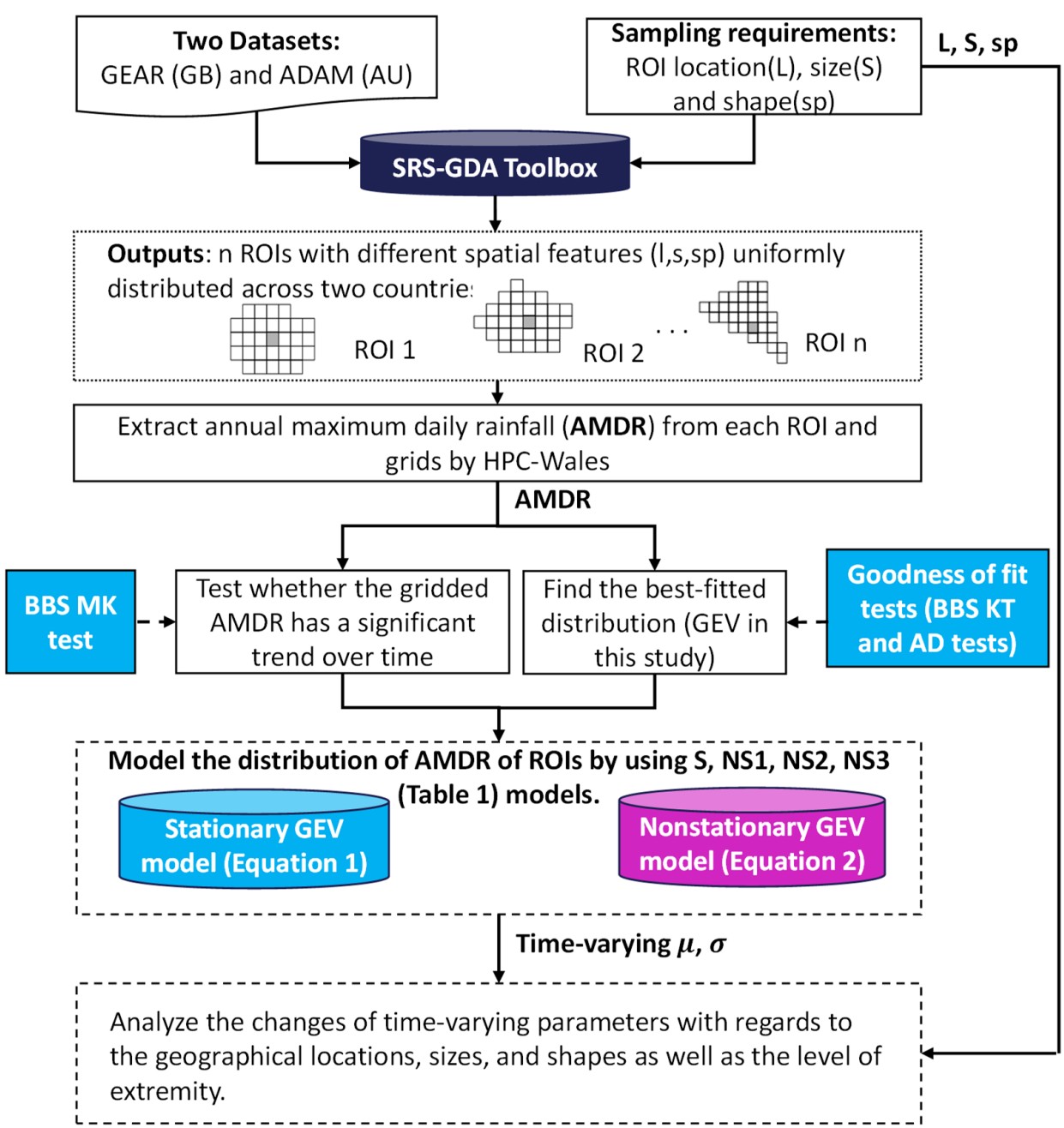

**Figure 1.** The methodology of the study.

*2.1. Data, ROIs and Extreme Rainfall in Two Countries*

The two century-long datasets, i.e., the 'Gridded Estimates of daily Areal Rainfall' (GEAR) and the 'Australian Data Archive for Meteorology' (ADAM) are applied. The GEAR dataset is a grid-based ($1 \times 1$ km$^2$) rainfall estimation covering the mainland of Great Britain (GB) from 1 January 1898 to 31 December 2010 projected onto the National Grid Reference coordinate system [24] with the easting ($x$-) and northing ($y$-) expressed in linear kilometres [25]. The ADAM dataset is also a grid-based ($0.05° \times 0.05°$, approx. $5 \times 5$ km$^2$) rainfall dataset from 1 January 1900 to 31 December 2018 over Australia (AU) based on the Geocentric Datum of Australia 1994 [26] with the origin of (44° S, 112° E) and easting ($x$-) and northing ($y$-) transformed to kilometres [27]. The daily rainfall amount of two datasets refers to the 24 h prior to the reporting time for the ADAM dataset and the 24 h after for the GEAR dataset, respectively.

To have an initial assessment of the spatio-temporal variation in the daily rainfall extremes, we employed the Block Bootstrapping Mann–Kendall (BBS-MK) trend analysis [28–30] to test whether the trend of the grid extreme rainfall is significant during the study periods at a significant level of 0.05. The magnitude of the trend, indicated by the Kendall's tau (see Figure 2a,b), shows that the majority of regions in the Midland of England, Wales and the Coastal regions of Scotland present a positive trend whilst western Australia shows an increasing trend but eastern regions show no or decreasing trend. This analysis provides the evidence for further detecting the possible nonstationarity in the daily rainfall and its strong association with geophysical locations, which moves forward from and complements our previous study [20] that focused only on the AMDR itself under the stationary assumption.

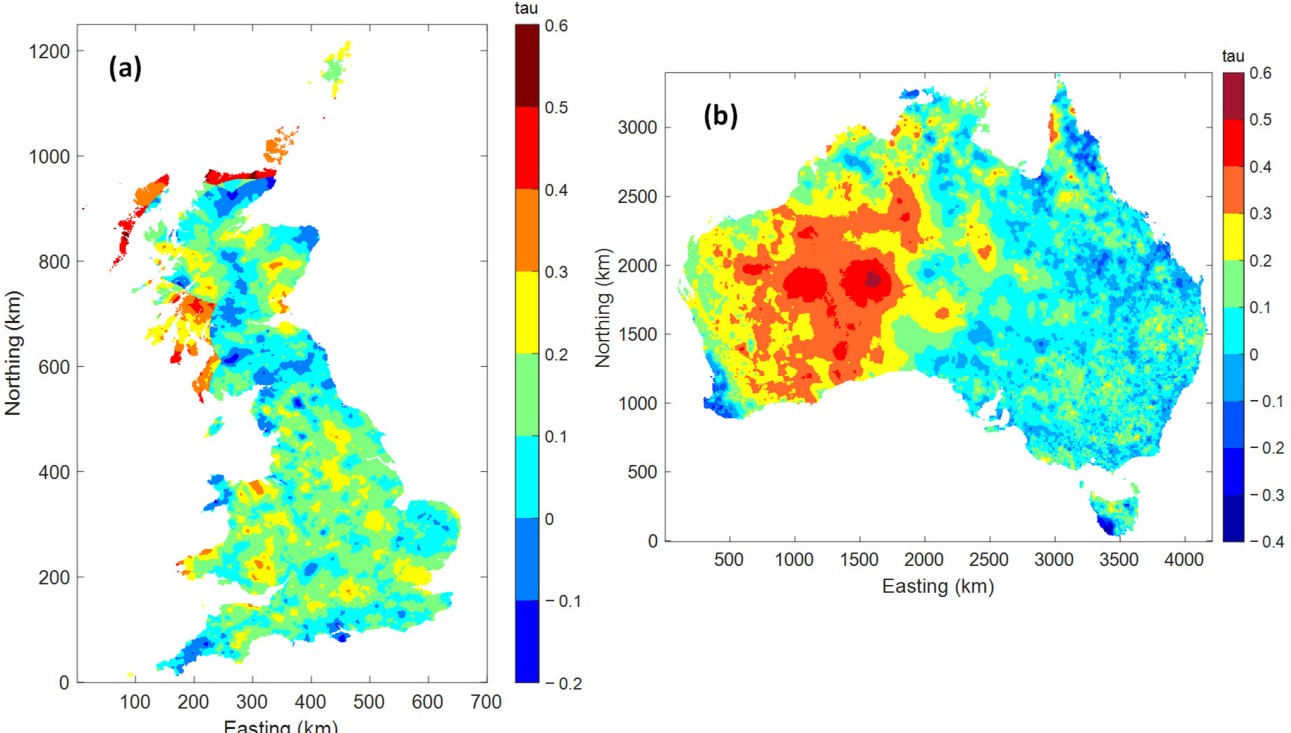

**Figure 2.** The trend of gridded AMDR over the last century in GB (**a**) and AU (**b**).

The spatial features of the sampled ROIs (see Figure 3a,b) generated by the SRS-GDA toolbox [31] follow the same setups in [20], uniformly positioned across two countries marked by their geometric centroids, randomized with sizes (10–1050 km$^2$ for GB case and 125–9900 km$^2$ for AU case) and shapes (controlled by the spatial index *sp* varying from elongated to rounded with east–west and north–south orientations). For each ROI, the areal daily rainfall is calculated by taking the arithmetic average before the ROI-orientated AMDR is generated. This procedure is carried out using the HPC-Wales resources due to the huge amount of data.

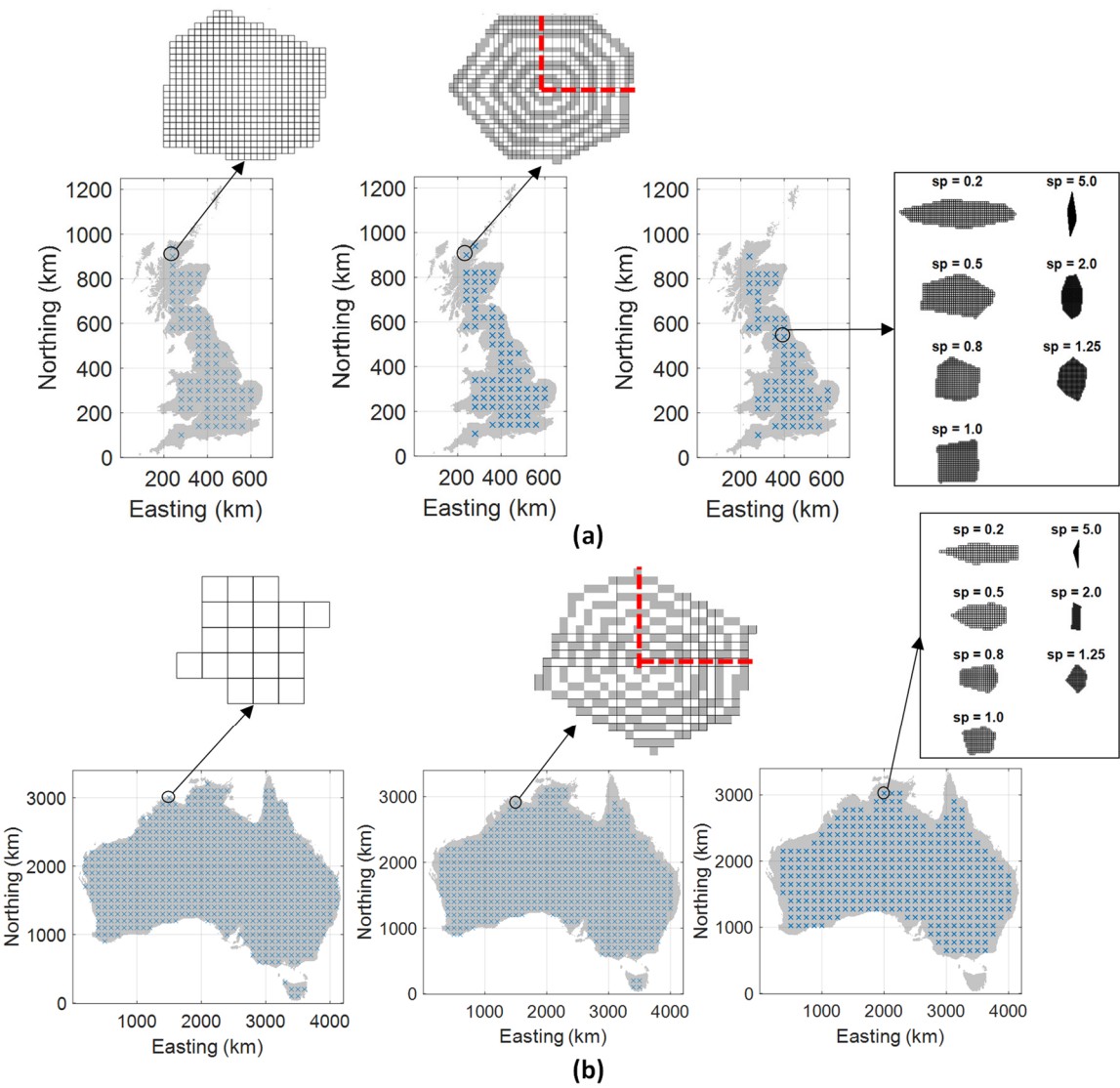

**Figure 3.** ROIs and their locations spatially distributed in GB (**a**) and AU (**b**) for this study: each subfigure can be divided into three panels where the left one demonstrates an ROI with the same size of 500 km² and a relatively rounded shape for the analysis in Section 3.1; the middle demonstrates a group of 10 ROIs which have a fixed shape but an incrementally increased size by defining an 20% increase in the main axis (marked as red dash line) in each iteration for the analysis in Section 3.2; the right panel depicts a group of 7 ROIs which have fixed size but varying shape described by spatial index *sp*. "×" marks the central location of these single or group of ROI(s) which are distributed uniformly over two countries. More details for generating ROIs can be checked in [20].

*2.2. Stationary and Nonstationary Generalised Extreme Value Models and Return Period*

The AMDR time series extracted from each ROI is then fitted by the GEV distribution whose cumulative distribution function (CDF) is defined as:

$$F(x; \sigma, \mu, \xi) = exp[-(1 + \xi(\frac{x - \mu}{\sigma}))^{-1/\xi}] \tag{1}$$

where $F$ is defined for $1 + \xi(x - \mu)/\sigma > 0$, $-\infty < \mu < \infty$, $\sigma > 0$ and $-\infty < \xi < \infty$; $\mu$, $\sigma$ are the location and scale parameters, $\xi$ is the shape parameter controlling the three limiting types of GEV: Gumbel ($\xi = 0$), Fréchet ($\xi > 0$) and the reversed Weibull ($\xi < 0$) distribution.

In the stationary model (S), three parameters are invariable with time or other covariates and estimated by the commonly used Maximum Likelihood method (ML). Several widely used candidate distributions were employed to fit the AMDR series and the goodness of fit was tested by using the Kolmogorov–Smirnov (KS) [32,33] and Anderson-Darling (AD) tests [34,35] at a significant level of 0.05, which shows that the GEV distribution performs best where more details can be seen in Wang and Xuan [20]. The GEV distribution fits well with the AMDR series of all ROIs with a 100% pass rate on the KS test and more than 97% on the AD test.

For nonstationary models (NS1–3), an additional subscript $t$ is added to GEV parameters which indicates that parameters are time-dependent; and the nonstationary CDF is:

$$F_t(x_t; \sigma_t, \mu_t, \xi) = \exp[-(1 + \xi(\frac{x_t - \mu_t}{\sigma_t}))^{-1/\xi}]　\quad (2)$$

To create a stable quantile estimation consistent with the behaviour of rainfall extremes, four different GEV models are developed with different assumptions regarding the parameters, as listed in Table 1.

**Table 1.** Stationary and nonstationary GEV models and the estimation methods used in this study.

| Description | Parameters | Estimation Method(s) |
|---|---|---|
| Stationary model: $F(x; \sigma_0, \mu_0, \xi)$ | $\sigma_0, \mu_0, \xi$ are constant | ML [1] |
| Nonstationary model 1: $F_t(x_t; \sigma_0, \mu_t, \xi)$ | $\mu_t = \mu_0 + \mu_1 \times t$ $\sigma_0, \xi$ are constant | ML and B-MCMC [2] |
| Nonstationary model 2: $F_t(x_t; \sigma_t, \mu_t, \xi)$ | $\sigma_t = \sigma_0 + \sigma_1 \times t$ $\mu_t = \mu_0 + \mu_1 \times t$ $\xi$ is constant | ML and B-MCMC |
| Nonstationary model 3: $F_t(x_t; \sigma_t, \mu_t, \xi)$ | $\sigma_t = \exp(\sigma_0 + \sigma_1 \times t)$ $\mu_t = \mu_0 + \mu_1 \times t$ $\xi$ is constant | ML and B-MCMC |

Note: [1] short for "Maximum Likelihood" method. [2] short for "Bayesian Markov-Chain Monte-Carlo" method.

Two parameter estimation methods, i.e., ML and the Bayesian Markov-Chain Monte-Carlo (B-MCMC) methods are employed to estimate the parameters of nonstationary models. Instead of obtaining the parameters directly, the B-MCMC method makes use of the Bayesian inference to estimate the posterior distribution of parameters based on prior knowledge. To make full use of the knowledge, the estimated parameters of stationary GEV were used to define the initial priori values of the nonstationary parameters drawn from uniform distribution sampling by the Latin Hypercube sampler [36]. Then, we broadly followed the algorithm presented in Sarhadi and Soulis [13] and Sadegh, Ragno [37] to carry out numerical iterations to explore the posterior distribution using the MCMC simulation with Metropolis and Gibbs sampling [38–40]. The calculated Metropolis ratio is used to accept or reject proposal status and the convergence of simulation is monitored by the Gelman-Rubin $\hat{R}$ diagnostic [41]. More details are provided in the Supplementary Text S1.

The criteria for selecting the best-fitted models include the root mean squared error (RMSE), the Akaike Information Criterion (AIC [42]) and the Bayesian Information Criterion (BIC [43]). Small values of these indexes indicate a better performance of the model in question (see Text S2).

To define the nonstationary return period, the time-varying nature of the nonstationary exceedance probabilities should be recognised, i.e., for a given threshold $x_0$, the number of expected exceedances over $T$ years is [44]:

$$K_t(x_0) = \int_0^T [1 - F_t(x_0)]dt　\quad (3)$$

For annual maxima over $N$ years, this leads to

$$K_N(x_0) = \sum_{i=1}^{N}[1 - F_i(x_0)] \times 1 \qquad (4)$$

where $F_i$ is the nonstationary CDF for the $i$th year. Correspondingly, the return period is $\tau_N(x_0) = N/K_N(x_0)$. Note that a subscript $N$ is used here to indicate the fact that both the return level and the expected number of exceedances are dependent on the duration (the $N$ years).

## 3. Results

### 3.1. Selection of Stationary and Nonstationary Models and Spatial Nonstationary Patterns

The best-fitted model of each ROI is selected by choosing the model with the smallest values of the criteria (RMSE, AIC and BIC). Results show that overall: (1) around 35% of ROIs in GB prefer stationary model while the rest 45% select NS1 (only $\mu$ is time-varying) and 20% select NS2 and NS3 (NS2–3; both $\mu$ and $\sigma$ are time-varying); (2) AU has a relatively lower ratio of ROIs favouring stationary model (around 20%) while 50% prefer NS1 and 30% prefer NS2–3. As to the methods used to fit the preferred nonstationary models, the ML method performs better for 60% of the GB cases compared with 40% performed by the B-MCMC method. In the AU cases, the ML method is significantly more dominating, e.g., with a ratio of 90% vs. 10%. More details are shown in Text S3.

The spatial distribution of model preference, i.e., stationary GEV versus nonstationary GEV, is further demonstrated in Figure 4a,e where ROIs with the same size of 500 km$^2$ and a relatively rounded shape is used. Geographically, those ROIs in GB that prefer nonstationary models are located along or near the coastal regions, especially in eastern and northern GB and the Scotland Highland. In AU, nonstationary models dominate the inland area and the majority of the south-western coastline while the north coastline of AU and the majority inland of the Northern Territory favour stationary model.

As to the types of the selected GEV, Figure 4b,f present the spatial variation in the GEV types of the best-fitted models of these ROIs where the majority follows the Fréchet distribution. Out of all ROIs in GB, there are nearly 80% that follow the Fréchet distribution, mainly located inland; around 16% follow the Weibull distribution located on the western coast. In AU, around 90% of ROIs follow the Fréchet distribution and only a very small proportion (3%) follows the Gumbel distribution (details in Table S2).

Figure 4c,d (UK case) and Figure 4g,h (AU case) reveal the time-varying changes of $\mu$ and $\sigma$ of the ROIs with their best-selected GEV model. If the ROIs prefer the stationary model where all parameters are constant, the rate of change in parameters is zero and marked as white colour. Additionally, for the ones preferring the nonstationary model, we calculated the rate of the change that is the difference between the parameter values at the end of the study period (i.e., 2010 for GB and 2018 for AU, $\mu_{end}$ or $\sigma_{end}$) and the starting times (i.e., 1898 for GB and 1990 for AU, $\mu_{start}$ or $\sigma_{start}$) divided by the values at the starting times (see Equation (5)).

$$rate(\mu) = (\mu_{end} - \mu_{start})/\mu_{start}$$
$$rate(\sigma) = (\sigma_{end} - \sigma_{start})/\sigma_{start} \qquad (5)$$

In GB, the changes of $\mu$ are in the range of ±10% and the ROIs with a decreasing $\mu$ are mainly located in southern Scotland and the regions between London and Birmingham while the majority of areas show a non-decreasing $\mu$ which indicates that the level of most frequent AMDR is non-decreasing. However, in AU, the south-middle zone and the eastern coasts are dominated by an increasing $\mu$ up to the rate of +20% while the north coast of Northern Territory and west-south coast of Western Australia are controlled by a decreasing $\mu$ with the rate of −5%. The majority of regions of GB and AU are observed to have a constant $\sigma$ while the rest region shows a decreasing $\sigma$ scattering near the coasts of England, which somehow indicates a decreasing occurrence probability of extremes. These changes are consistent with the BBS-MK trend analysis as presented in Figure 2.

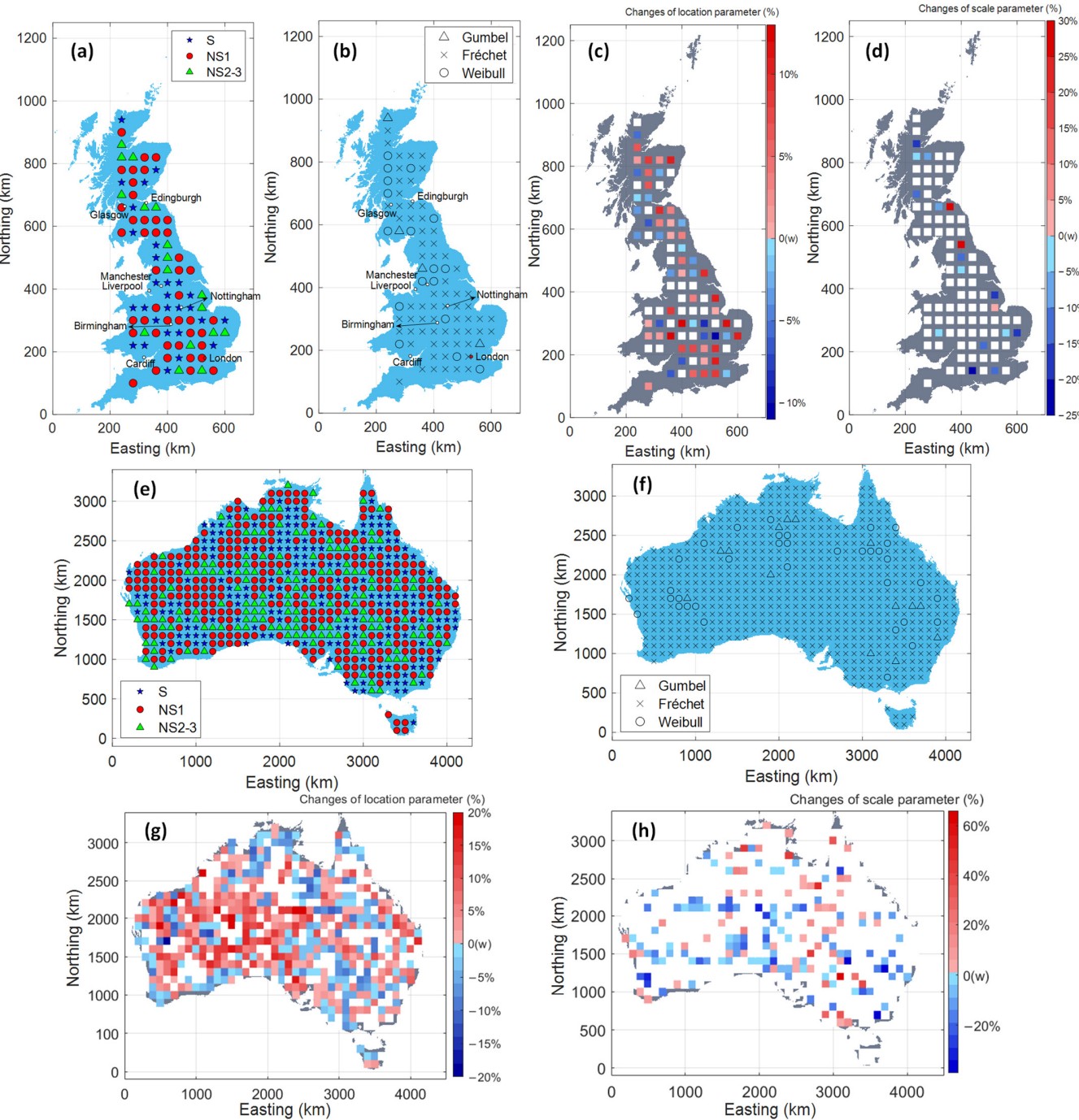

**Figure 4.** Spatial distribution of ROIs with the size of 500 km² and relative rounded shape in terms of (1) the best-selected model type in GB (**a**) and AU (**e**); (2) the best-fitted GEV type in GB (**b**) and AU (**d**); (3) the changes of location (**c,g**) and scale parameters (**f,h**) in percentage within the record periods (113 years for GB and 129 years for AU) and please noted that white colour ("w" specified in the colour bar) indicates the ROI with no change in the parameters.

### 3.2. Spatial Variation in Nonstationary Patterns over ROI Size

In GB, the proportion of ROIs that prefer a stationary model gradually increases as the ROI size increases (60% for ROI size < 100 km²; 65% for ROI size in 100 km² ~ 500 km² and 67% for ROI size > 500 km²). However, such proportion in AU is relatively stable and stays around 25% for stationary model and 75% nonstationary models, regardless of the ROI size (see Table S1).

To help the discussion, we introduce the rate of the change in GEV parameters associated with the incremental change in ROI size (i.e., +20% each), denoted as $\Delta$. $\Delta$ is the Sen'slope [45] of the BBS-MK test applied to detect the changes of parameters over region size at the significant level of 0.05 and the white colour ($\Delta = 0$) in Figure 5 indicates the insignificant change. With the increase in ROI size, the reddish colour represents a positive $\Delta$ which means the parameter increases as well, while the bluish indicates the negative case. Four parameters for analysis are $\mu_0$ and $\sigma_0$ which indicate the baselines, i.e., the estimation of the baseline climate without taking climate change impact into consideration, referring to the level of most frequent AMDR and the occurrence probability of extremes; and $\mu_1$ and $\sigma_1$ which are the time-varying changes from such baselines due to climate change while equal to zero if the best-selected model is stationary.

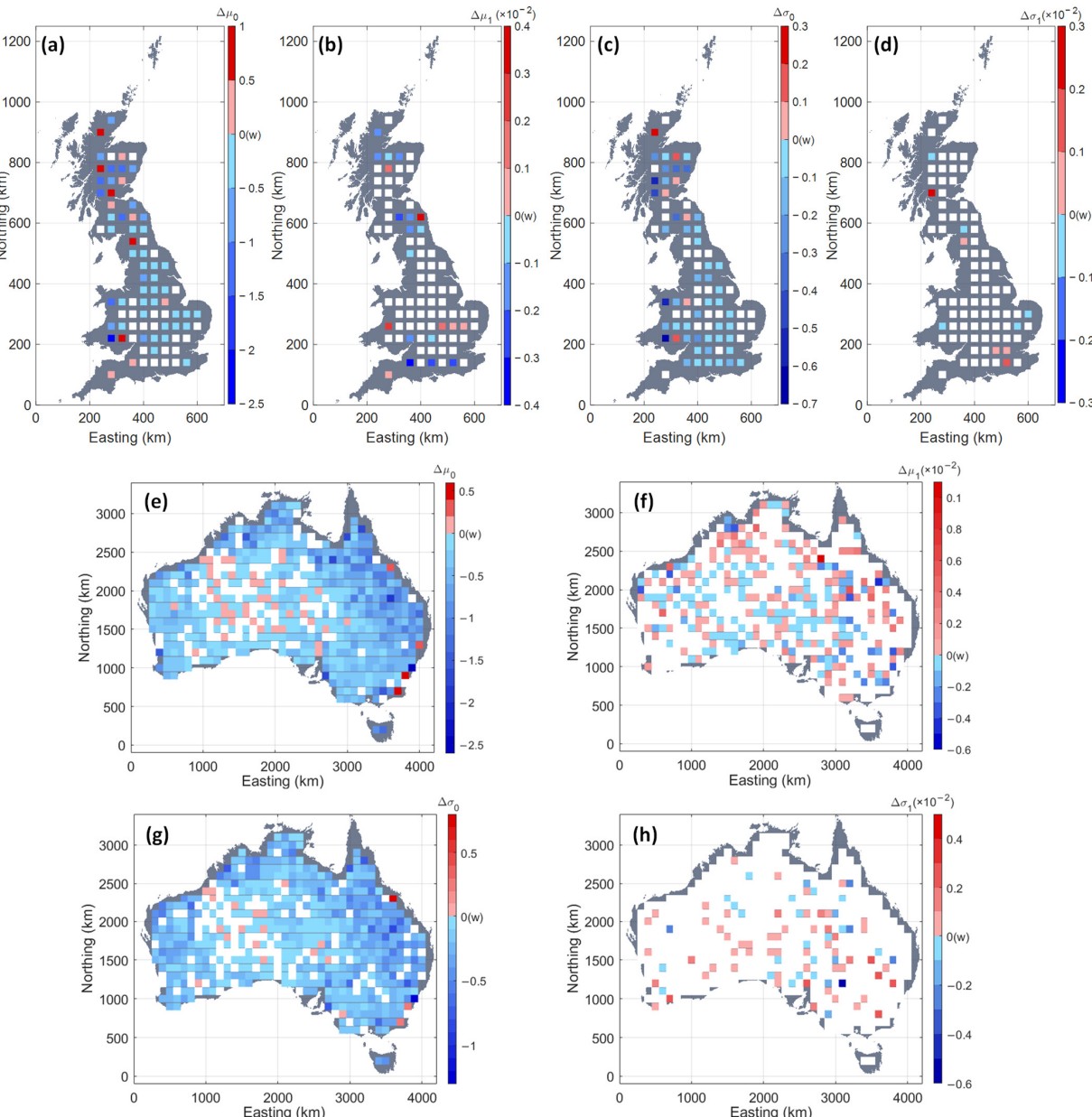

**Figure 5.** Spatial distribution of ROI groups whose parameters ($\mu_0, \mu_1, \sigma_0, \sigma_1$) change with the increase in ROI size in GB (**a**–**d**) and AU (**e**–**h**). Please noted that the white colour indicates the ROI with an insignificant change in the parameters. Noted that "w" shown in the colour bar indicates the colour white which means the changes of the parameter is zero.

In both countries (Figure 5a,c,e,g), most regions show decreasing baselines ($\mu_0$ and $\sigma_0$) with the increase in ROI size, especially in AU. Such a decrease can be attributed to the areal average when involving more grids in the sampled ROI. However, near the coastal regions of GB and the boundary of the south-middle dry zone of AU, some ROIs have an increased baseline because the increasing size will involve more grids of higher rainfall which may overcompensate the reduction caused by the areal average.

In addition, the time-varying terms ($\mu_1$ and $\sigma_1$, see Figure 5b,d) are insignificant in most of GB while very few with significant change locate near the coasts and have a decreasing $\mu_1$. In AU (Figure 5f), regions with a decreased $\mu_1$ are mainly located in the middle-south zone while the others with an increased $\mu_1$ are closer to the coasts. However, the spatial distribution of the changes of $\sigma_1$ is more random than $\mu_1$ and most regions present an insignificant trend of $\sigma_1$ in both countries.

The time-varying terms $\mu_1$ and $\sigma_1$ can reflect how the most frequent AMDR and the occurrence probability of extremes change over time affected by climate change in the last century. Interestingly, such impact is not always coincident with the decreased average baseline climate estimation ($\mu_0$ and $\sigma_0$) due to the statistical average of larger ROI sizes but influenced more by the geographical locations. For example, for the ROIs near the coasts in both countries, increasing their size can lead to an amplification of climate change impact on the most frequent AMDR and a higher probability for extremes to occur. However, in general, increasing region size will decrease both the average status of climate and climate change impact.

*3.3. Spatial Variation in Nonstationary Patterns over ROI Shape*

Figure 6 presents the changes in baselines ($\mu_0$, $\sigma_0$) and time-varying terms ($\mu_1$, $\sigma_1$) of the ROIs in GB and AU, parameterised by the ROI shape ($sp$) varying from an elongated west–east orientated ($sp = 0.2$, 0.5), gradually to more rounded ($sp = 0.8$, 1.0, 1.25), then to an elongated but north–south orientated ($sp = 2.0$, 5.0). A small difference is observed between the baseline parameters of ROIs with reciprocal shape indexes especially in AU, regarding a symmetric pattern around $sp = 1.0$. Additionally, in both countries, most ROIs with a relatively rounded shape have higher values of the most frequent AMDR and the occurrence probability of extremes than those of elongated ones, referring to the distribution parameters $\mu$ and $\sigma$. In other words, compared with the analysis of temporal changes in extreme daily rainfall in the study areas with a more elongated shape in these two countries, the area with a relatively rounded shape is more likely to capture the temporal variation in extremes. However, it should be noted that the temporal variability in these parameters with different shapes of ROIs is insignificant compared with the changes with locations and ROIs' sizes.

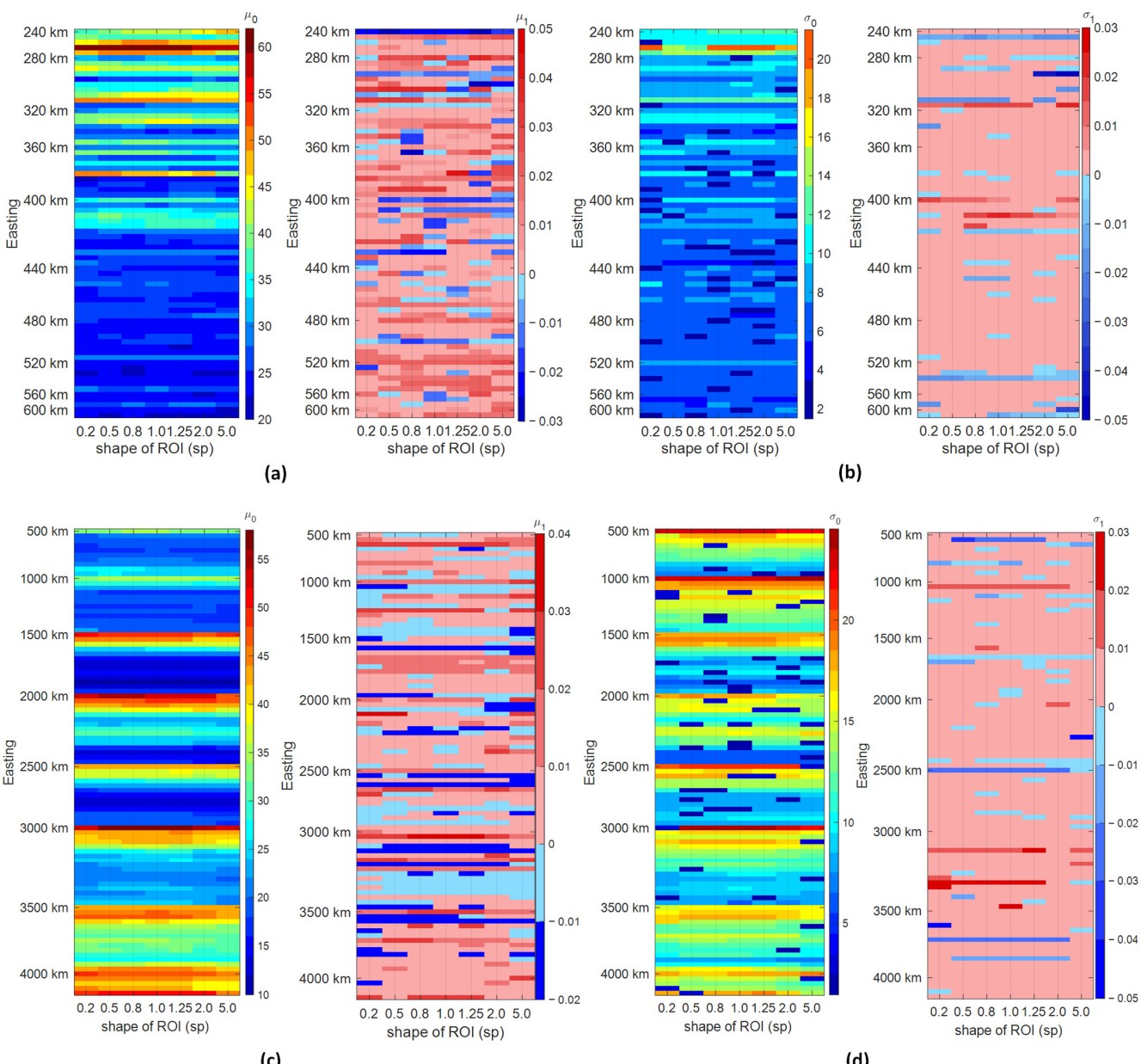

**Figure 6.** Both baseline and time-varying parameters ($\mu_0, \mu_1, \sigma_0, \sigma_1$) change over the ROI shape indicated by the shape index of *sp* in GB (**a**,**b**) and AU (**c**,**d**). The vertical axis indicates the location of the groups of ROIs (e.g., in Figure 6a, 10 groups (those at 240 km, 280 km, 320 km, 360 km, 400 km, 440 km, 480 km, 520 km, 560 km and 600 km) are presented from west to east while in each group, ROIs are presented from south to north (e.g., 5 ROIs presented between 240 km and 280 km are all geographically located at 240 km Easting from 580 km Northing to 900 km Northing)) and the colour bar shows the values of parameters. Due to the limited length of figures, the completed AU case with 2646 ROIs is shown in Supplementary Figure S1.

*3.4. Implication of Return Period*

　　To demonstrate the difference between the nonstationary and stationary models, 6 reference return levels of $x_0$ are selected, i.e., $\text{AMDR}_{\text{ref2}}$, $\text{AMDR}_{\text{ref5}}$, $\text{AMDR}_{\text{ref10}}$, $\text{AMDR}_{\text{ref25}}$, $\text{AMDR}_{\text{ref50}}$ and $\text{AMDR}_{\text{ref100}}$ corresponding to the return periods of 2, 5, 10, 25, 50 and 100 years, and calculated from the stationary model. Then, these stationary return levels are applied to calculate their corresponding return periods under the nonstationary condition (Equations (3) and (4)). The difference is then quantified between the nonstationary return

periods (NS-RP) and the referenced stationary return periods (S-RP) for the same given reference return levels.

Figure 7a,b present the NS-RP estimated by their best-selected nonstationary model for the three return levels $x_0$ from low to high (i.e., AMDR$_{\text{ref5}}$, AMDR$_{\text{ref25}}$ and AMDR$_{\text{ref50}}$) in GB and AU. In both countries, the difference between NS-RP and S-RP can be ignored (marked as white colour) at the lower return level (e.g., AMDR$_{\text{ref5}}$). It means that for estimating rainfall at a lower return level, the stationary model is able to capture the magnitude of rainstorms. However, at higher return levels, the percentage of ROIs with almost zero difference (marked as white) is decreased. In GB, higher return levels (e.g., AMDR$_{\text{ref25}}$ and AMDR$_{\text{ref50}}$) in North Wales, middle Scottish Highland and eastern and southern England (shown by blueish symbols) are underestimated by the stationary model while the coastal regions of Scotland and Wales witness an overestimation. In AU, the middle region of New South Wales and the coastal region of North Territory are shown an underestimation of higher return levels by the stationary model while the coastal area of Queensland and most regions of western Australia show an overestimation.

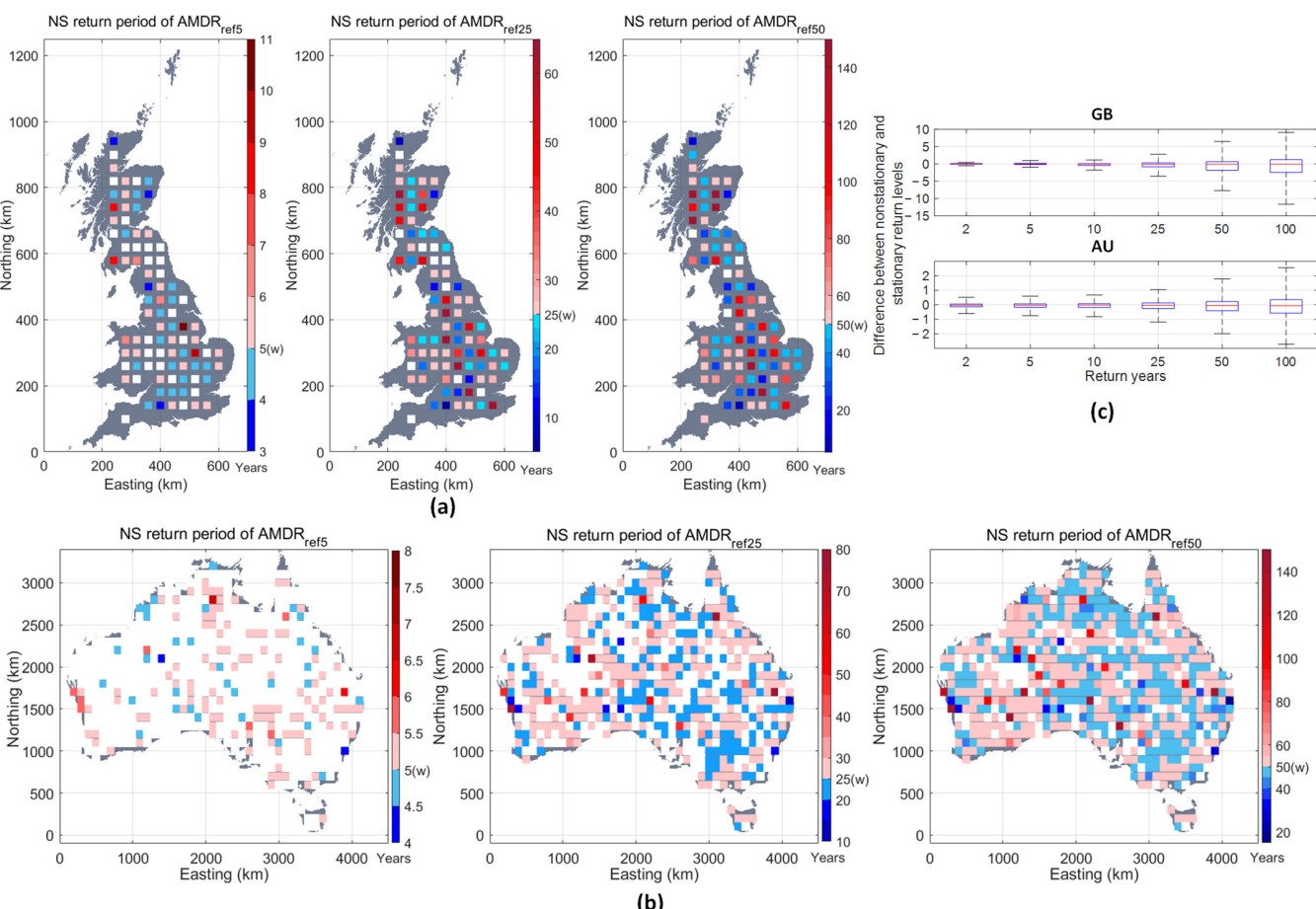

**Figure 7.** Nonstationary return periods corresponding to the return levels estimated by the stationary model and the spatial distribution referring to 1-in-5 years, 1-in-25 years and 1-in-50 years in GB (**a**) and AU (**b**). An overall comparison between the nonstationary and stationary return levels corresponding to the same return year is presented as a boxplot in (**c**) where the upper panel shows the GB case and the lower shows the AU case. Noted that "w" in the colour bar indicates the colour white which means there is no difference between stationary and nonstationary return periods.

Generally, with an increase in the return periods, the uncertainty that arises from the difference between the return levels calculated by the nonstationary and stationary model grows larger (see the length of the body and whiskers of each boxplot in Figure 7c). In GB,

a left-skewed boxplot is observed at the higher return periods, which means that the return levels in most ROIs estimated by a stationary model are higher than the corresponding nonstationary model estimate; while in AU, the difference is small and randomly distributed around zero and a half for overestimation and half for underestimation.

Combining the results with the best-selected models presented in Section 3.1, we can see that in GB, the AMDR in coastal regions, most parts of Scotland and the east–south of England, the nonstationary condition is preferred and underestimated by the stationary model, e.g., 1-in-50-year rainfall becomes 1-in-30 year estimated by the best-fitted nonstationary model. In AU the same situation happens in the inland areas which are fitted better by the nonstationary model, such as the middle region of New South Wales and the boundary area with Queensland, north coastal regions of North Territory, but the underestimation is small, e.g., 1-in-50 years becomes 1-in-45 years. This demonstrates that the currently used stationary model can result in high uncertainty and miss rare but high-risk extreme event occurrences, especially for the regions where the nonstationary condition is preferred, which leads to infrastructure damage.

## 4. Conclusions

This paper presents an analysis of the spatial variation in both the stationary and nonstationary GEV modelling of annual maximum daily rainfall (AMDR) extracted from over 11,010 regions of interest (ROI) with different spatial properties (location, size and shape) in Great Britain (GB) and Australia (AU) using two century-long grid-based datasets. Three nonstationary models with different time-varying GEV parameters ($\mu$ and $\sigma$) schemes are proposed and fitted using both the Maximum Likelihood (ML) and the Bayesian Markov-Chain Monte-Carlo (B-MCMC) methods, before being compared with the stationary model. Finally, the spatial patterns of the AMDR in both countries are analysed and quantified, with respect to the ROI's location, size and shape as well as the time-varying changes due to climate change. The following conclusions can be drawn:

(1) In general, the majority of the ROIs in both countries favour the nonstationary GEV model (NS-GEV) and most of them prefer the condition that only $\mu$ assumed to be linearly changing with time; most NS-GEV applications show the ML method performs better than the B-MCMC method (60% and 90% in GB and AU). AMDR of over 80% ROIs in both countries follows Fréchet distribution.

(2) Geographic location is the most significant factor that affects not only the average status of the baseline climate (with respect to $\mu_0$ and $\sigma_0$) but also the time-varying changes due to climate change (with respect to $\mu_1$ and $\sigma_1$). During the last century in GB, the changes in the level of the most frequent AMDR (with respect to $\mu$) are in the range of $\pm 10\%$ and the majority of areas show a non-decreasing trend. However, in AU, the south-middle zone and the eastern coasts are dominated by an increasing $\mu$ up to the rate of +20% while the north coast of Northern Territory and west-south coast of Western Australia are controlled by a decreasing $\mu$ with the rate of $-5\%$. The majority of regions of GB and AU are observed to a still $\sigma$ while some specific regions with a decreasing $\sigma$ scattering near the coasts of England indicate a decreasing occurrence probability of extremes.

(3) Region size is shown to be a secondary factor. Generally, the two countries show a decreased average status of climate with an increase in size because of the statistical average. However, near the coastal regions of GB and the boundary of the south-middle dry zone of AU, some ROIs have an increasing status. Although the effect of region size on time-varying changes is insignificant, the climate change impact is not always decreased with the increase in region size but is influenced by geographical locations.

(4) Region shape does not significantly affect either the average climate status or time-varying changes; however, a symmetric pattern of average climate status is found for regions with reciprocal spatial indexes and the average climate status in ROIs with a relatively rounded shape is usually higher than the elongated ones

(5)   The stationary GEV models underestimate the risk in several specific regions such as the coastal regions in both countries where the nonstationary model is preferred. It may inspire a re-consideration of the current design storm determination procedure.

We trust that the findings from this study are valuable for the civil engineering community in a way that not only do they further corroborate other research findings on extreme rainfall, e.g., extremes are likely to become more frequent due to climate change impact, but they also quantitatively address how such changes over not only climate but also the geographical location, size and shape may affect the prevailing engineering design standard.

Further work is recommended to investigate closely the underlying datasets with respect to potential inconsistency in the resolution of the data observed near the West coast of Scotland and the AU coasts. In addition, a comparative study with long-term, single gauge observations, as well as catchment-orientated sampling is likely to make more robust conclusions. As the changes of rainfall extremes are very likely to be affected by many other climate indices such as temperature, pressure, El Niño-Southern Oscillation (ENSO), etc., further work is recommended to also include these related indices as covariates and to explore more possible covariate-varying patterns where the distribution parameters may follow.

**Supplementary Materials:** The following supporting information can be downloaded at: https://www.mdpi.com/article/10.3390/w15010128/s1, Text S1: Bayesian Markov-Chain Monte-Carlo (BMCMC) Method; Text S2: Criteria for selecting best-fitted models; Text S3: Selection of stationary and nonstationary models and spatial patterns; Figure S1: Both baseline and time-varying parameters ($\mu_0, \mu_1, \sigma_0, \sigma_1$) change over the ROI shape indicated by the index of $sp$ in AU. The horizontal axis indicates the location index the ROIs and colour bar shows the values of parameters; Table S1: the number of ROIs with their best-fitted model varying with regional sizes and shapes in GB and AU; Table S2: the number of ROIs with their best-fitted model varying with regional sizes and shapes in GB and AU. References [31,37,41,46–49] are cited in the supplementary materials.

**Author Contributions:** Conceptualization, H.W. and Y.X.; methodology, H.W. and Y.X.; validation, H.W.; formal analysis, H.W. and Y.X.; writing—original draft preparation, H.W.; writing—review and editing, Y.X.; supervision, Y.X.; project administration, Y.X.; funding acquisition, Y.X. and H.W. All authors have read and agreed to the published version of the manuscript.

**Funding:** This research was funded by the Academy of Medical Sciences, Grant Ref: GCRFNGR4_1165.

**Data Availability Statement:** Two datasets applied in this study are available in the public domain online at https://doi.org/10.5285/33604ea0-c238-4488-813d-0ad9ab7c51ca (accessed on 27 December 2022) for the GEAR dataset and http://www.bom.gov.au/jsp/awap/rain/index.jsp for the ADAM dataset (access on 27 December 2022).

**Acknowledgments:** The authors would like to thank the Centre of Hydrology and Ecology (CEH) and The Bureau of Meteorology, Australia for providing the datasets. The open-source toolbox of spatial random sampling for grid-based data analysis (SRS-GDA toolbox, http://doi.org/10.5281/zenodo.4044626 accessed on 27 December 2022) developed by the authors used in this study. This research is supported by the Chinese Scholarship Council and the College of Engineering, Swansea University, UK via their PhD scholarships offered to the co-author Han Wang and the Academy of Medical Sciences GCRF Networking Grant (REF: GCRFNGR4_1165) which are gratefully acknowledged.

**Conflicts of Interest:** The authors declare no conflict of interest.

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
