# Peer review of "An Area-Orientated Analysis of the Temporal Variation of Extreme Daily Rainfall in Great Britain and Australia"

_water, doi:10.3390/w15010128_

Round 1
Reviewer 1 Report
This paper deals with non-stationary extreme precipitation analysis in Great Britain and Australia with scientifically sound results which could be of interest for the scientific community and the large audience of this journal. However, some aspects indicated below could contribute to improve the paper content and quality.
- L145-150: if the stationary model was found to fit so well with the AMDR series of all ROIs with a 100% pass on the KS test and more than 97% on the AD test according to ref 20, how come that the non-stationary model overpass the stationary model in the latter development? Are the trend results not in contradiction with this successfulness of the stationary model indicated above?
- L153-155: Why the used covariate was limited only to time as the authors indicated that there are other covariates? why other covariates were not explored such as climate indices? Even, limiting the covariate to time, there are more options that could be explored such as multiple polynomial functions, etc.
- L203-204: How were the change of location and scale parameters calculated here? Considering the extreme values may not be the best option. Why not computing the Sen slopes after splitting the data into two parts and then deduce the percentage change?
- L233: Should be ‘Sen 'slope’ not ‘Sen 'slop’
- L267: Could not really see the usefulness or added values of this section. (Spatial variation of nonstationary patterns over ROI shape). Please, clarify.
- L293: ‘However, for a higher return level, the difference becomes significant’: Did the authors apply a statistical test to find significant changes? The difference is not too high in absolute value mainly for AU if we consider the interquartile ranges. Anyhow, a test should be applied.
Author Response
Please find the authors' response in the file attached.

Reviewer 2 Report
It is avery interesting paper and the authors have made a great work. I sugesst to be accept as is except if the authors want to make any English revisions.
Author Response
Many thanks for your time spending in reviewing our paper. We have gone through the MS carefully to check and correct any typos or English mistakes.
Reviewer 3 Report
This manuscript employed different techniques to gain insight characteristics of annual maximum daily rainfall over two large countries, Great Britain and Australia. The method sounds reasonable; the workload is large, results are interesting and significant to readers. I do not understand the reason for selecting 500 km2 for ROIS in both studied regions. Any justification for the selection of this size and description of upscaling from original datasets to 500 km2 would make the manuscript clearer to readers. I recommend a publication after a minor revision.
Author Response

(The authors gave the same response as above.)

Reviewer 4 Report
1. Provide the procedure diagram to improve the readablity. The steps provided in section 2 is not sufficient.
2. Figure1(b and d) change the scale into real instead *5 (also the other figures accordingly). Same scale color bar for (a) and (b) might be more reasonable.
3. Figure 1 can be separated into 2 figures
Figure 1(a) and (b)
Figure 1(c) and (d) -->figure 2.
Further detail description can be made for Figure1(c) and (d).
I cannot follow the caption description yet.
3. Figure 2 (b) and (f) why only 3 distributions. Not compared with GEV itself?
4. Check Eq.4 and also cite a reference.
5. Color bar scale is different each panel. Same color bar for the same statistics should be more meaningful.
6. sp --> shape (better with full name)
Author Response

(The authors gave the same response as above.)
